# Phloretin Transfersomes for Transdermal Delivery: Design, Optimization, and In Vivo Evaluation

**DOI:** 10.3390/molecules28196790

**Published:** 2023-09-24

**Authors:** Jiawen Wang, Yuanyuan Zhao, Bingtao Zhai, Jiangxue Cheng, Jing Sun, Xiaofei Zhang, Dongyan Guo

**Affiliations:** 1State Key Laboratory of Research & Development of Characteristic Qin Medicine Resources (Cultivation), and Shaanxi Key Laboratory of Chinese Medicine Fundamentals and New Drugs Research, and Shaanxi Collaborative Innovation Center of Chinese Medicinal Resources Industrialization, Shaanxi University of Chinese Medicine, Xi’an 712046, China; nuyoah1119@163.com (J.W.); zbp@sntcm.edu.cn (B.Z.); cjx511@sntcm.edu.cn (J.C.); ph.175@163.com (J.S.); 2051028@sntcm.edu.cn (X.Z.); 2Yulin Hospital of Traditional Chinese Medicine, Yulin 719000, China; zhaoyuanyuan48@163.com

**Keywords:** phloretin, Phl-TFs, transdermal administration, PTG, pharmacokinetics, antioxidant

## Abstract

Background: Phloretin (Phl) is a flavonoid compound that contains multiple phenolic hydroxyl groups. It is found in many plants, such as apple leaves, lychee pericarp, and begonia, and has various biological activities, such as antioxidant and anticancer effects. The strong hydrogen bonding between Phl molecules results in poor water solubility and low bioavailability, and thus the scope of the clinical application of Phl is limited. Therefore, it is particularly important to improve the water solubility of Phl for its use to further combat or alleviate skin aging and oxidative damage and develop antioxidant products for the skin. The purpose of this study was to develop and evaluate a phloretin transfersome gel (PTG) preparation for transdermal drug delivery to improve the bioavailability of the drug and delay aging. Methods: Phloretin transfersomes (Phl-TFs) were prepared and optimized by the thin-film dispersion–ultrasonication method. Phl-TFs were characterized by transmission electron microscopy (TEM), differential scanning calorimetry (DSC), Fourier transform infrared (FTIR) spectroscopy, and X-ray diffraction (XRD). The Log P method was used to determine the solubility of the Phl-TFs. The skin penetration ability of the prepared PTG was evaluated using the Franz diffusion cell method. In addition, the in vivo pharmacokinetics of PTG were studied in rats, and an antioxidant activity investigation was conducted using a D-gal rat model. Results: Phl-TFs were successfully prepared with a Soybean Phosphatidylcholine (SPC)/CHOL ratio of 2.7:1 *w*/*v*, a phloretin concentration of 1.3 mg/mL, a hydration time of 46 min, an ultrasound time of 5 min, and an ultrasound power of 180 W. The Log P was 2.26, which was significantly higher than that of phloretin (*p* < 0.05, paired *t* test). The results of the in vitro penetration test demonstrated that the cumulative skin penetration of the Phl-TFs after 24 h was 842.73 ± 20.86 μg/cm^2^. The data from an in vivo pharmacokinetic study showed that the C_max_ and AUC of PTG were 1.39- and 1.97-fold higher than those of the phloretin solution gel (PSG), respectively (*p* < 0.05, paired *t* test). The experimental results in aging rats showed that PTG had a better antioxidant effect. Conclusions: Phl-TFs and PTG preparations with a good shape, safety, and stability were successfully prepared. In vivo pharmacokinetics and preliminary antioxidant experiments further verified the transdermal penetration and antioxidant activity of the phloretin transdermal drug delivery preparation, providing an experimental basis for its further development.

## 1. Introduction

Aging is a systemic condition that has been of significant concern. Skin aging is the most intuitive expression of this phenomenon and the most direct and favored model for researching aging mechanisms [1,2]. Skin aging is a complicated, multidimensional process that can be categorized as endogenous and exogenous. Endogenous aging mostly considers normal processes that occur with age. Moisturizing is essential to prevent the natural aging of the skin because dry skin and dehydration are major factors in this process. Exogeneous aging is a controlled type of skin aging caused by environmental factors such as UV radiation, cigarette smoke, wind, and sunlight, which leads to skin thinning, collagen loss, and the degradation of elastic fibers [3]. The primary cause of external skin aging is exposure to UV radiation, which can cause skin damage and redness. It is known as photoaging when these variables lead to skin aging. Photoaging accelerates skin aging much more quickly than natural aging. Photoaging is characterized by a rough, dry appearance as well as creases and sagging [4]. Aging reduces the body’s capacity to scavenge free radicals due to the reduced activity of associated antioxidants, including GPx and SOD [5]. The levels of reactive oxygen species (ROS) can increase due to various factors, such as mitochondrial damage and the inflammatory response. These factors can damage neuronal cell membranes by creating lipid peroxides from unsaturated fatty acids, which speeds up the process of functional aging and associated age-related diseases such as Alzheimer’s disease and Parkinson’s disease [6]. Natural skin aging is the result of increased oxidative stress resulting from the decline in the cellular repair capacity that occurs with age [7].

Phl, also known as trihydroxy phenol acetone, is a flavonoid found mainly in the peel and root bark of juicy fruits such as apples and pears. Researchers are exploring the significant biological effects of Phl on human health, including its antioxidant, memory-improving, and anticancer functions. Phl possesses skin-lightening and anti-aging properties, making it a promising compound for developing new pharmaceuticals and natural health foods [8,9,10]. However, the clinical use of phloretin is restricted by first-pass metabolism, poor oral absorption, and an absolute bioavailability of 8.68% [11]. Thus, a distinctive system must be devised for its delivery.

Transdermal administration can significantly reduce the first-pass effect, minimize gastrointestinal irritation, facilitate a consistent rate of transdermal absorption into the bloodstream, and enhance long-term blood concentration stability [12]. One of the major obstacles in drug absorption through the skin’s outermost layer, the stratum corneum, is the stratum corneum itself. One of the latest trends in formulation research is utilizing advanced formulation technologies instead of permeation promoters to assist with the transdermal absorption of medications [13]. Transfersomes (TFs) are delivery vehicles that require dermal penetration for their effects to occur.

TFs differ from traditional liposomes in that they are prepared by incorporating different edge activators into the phospholipid component with high deformability, differential pressure permeability, strong skin affinity, and high safety [14,15]. TFs also offer the benefits of amphiphilicity and targeted modification. With enormous potential for transdermal drug delivery, TFs can encase insoluble pharmaceuticals in a bilayer, pass through difficult-to-cross barriers, reach the intended site, release the drug in a slow, controlled manner, and offer therapeutic effects. Miatmoko et al. found that transfersome-loaded AMSC-MP was safe and effective, improving dermal delivery and delaying skin aging [16]. Ming Yuan et al. encapsulated indomethacin (IND) in HA-modified transfersomes (IND-HT) to improve the transdermal delivery of IND and minimize adverse effects, and the results of their study suggest that the developed topically administered IND-HTs/Gel could offer a promising alternative to oral IND delivery [17]. These data suggest that transfersomes improve the controlled release of drugs in transdermal delivery systems, supporting previous research [18].

In this study, the safety of various Phl-TF formulations that had been optimized in terms of their physical characterization, oil–water partition coefficient, in vivo skin permeability, in vivo pharmacokinetics in rats, and in vivo anti-aging benefits was examined. This study aims to establish an experimental foundation for creating phl transdermal formulations to attenuate the aging process in living organisms.

## 2. Results

### 2.1. Preparation and Optimization of Phl-TFs

#### 2.1.1. Single-Factor Experiments

##### Ratio of SPC to CHOL

The HPLC investigations strictly adhered to all requirements outlined in Appendix A and Appendix A. The molar ratios of SPC and CHOL were tested at 1:1, 2:1, 4:1, 6:1, 8:1, and 10:1, and the resulting encapsulation efficiency (EE) values were (84.20 ± 1.03)%, (94.06 ± 1.20)%, (93.90 ± 1.54)%, (91.50 ± 1.01)%, (90.90 ± 1.30)%, and (89.30 ± 1.05)%, respectively. Therefore, a ratio of SPC to CHOL of 2:1 was selected, as it resulted in the highest EE (Appendix A(1)).

##### Phloretin Concentration

The concentrations of phloretin evaluated were 0.5, 1, 2, 3, 4, and 5 mg/mL, and the EE values were (84.20 ± 1.05)%, (94.06 ± 1.02)%, (91.70 ± 1.03)%, (89.60 ± 1.04)%, (87.43 ± 1.03)%, and (80.48 ± 1.04)%, respectively. Thus, the maximum EE was achieved at a phloretin concentration of 1 mg/mL (Appendix A(2)).

##### Hydration Time

The ratio of SPC to CHOL was fixed at 2:1, and the compound was prepared for 15, 30, 45, 60, 75, and 90 min of hydration. The EE values under these conditions were (67.38 ± 1.06)%, (82.94 ± 1.03)%, (91.21 ± 1.04)%, (89.61 ± 1.04)%, (88.36 ± 1.02)%, and (86.57 ± 1.04)%, respectively. The EE first increased and then decreased with increasing hydration time, and 45 min was selected as the optimal time (Appendix A(3)).

#### 2.1.2. Phl-TFs Optimized by BBD-RSM

The experiment was designed using Design Expert 8.0.6 (Appendix A), and ANOVA is shown in Appendix A. The quadratic polynomial regression equation obtained was EE% = 94.62 + 0.22 A + 1.50 B + 0.30 C + 0.89 AB + 0.87 AC − 0.88 BC − 1.65 A^2^ − 7.69 B^2^ − 2.87 C^2^. After ANOVA, a value of *p* < 0.05 was obtained, indicating that the model is significant. The R^2^ of the model (0.8796) indicates that the regression model simulates the true conditions well and that the errors between the model and the test values are small.

According to the optimal process predicted using Design Expert (8.0.6) and the experimental data, a verification test was carried out. The optimal conditions were an SPC/CHOL ratio of 2.7:1 *w*/*v*, a phloretin concentration of 1.3 mg/mL, and a hydration time of 46 min, giving a theoretical EE of 94.72%. Three groups of transmitters were prepared in parallel, and the EE values were 93.80%, 95.36%, and 95.71%, respectively, with an average value of 94.96% and RSD of 1.01%, which verified the validity and reliability of the RSM model.

Design Expert 8.0.6 was used to predict the optimal conditions for the model, and the contour plot of the encapsulation rate and the 3D surface plot were obtained by fixing one factor and changing the other two factors (Appendix A(4)).

#### 2.1.3. Factorial Design to Optimize the Phl-TF Preparation

Factorial design was carried out according to the results of the single-factor experiment as shown in Appendix A. According to the results from the analytical design (Appendix A), the homogeneity and stability of the Phl-TFs were good, giving a maximum OD value when the ultrasonic time was 5 min and the power was 180 W. The average EE was (88.02 ± 1.92)%, the particle size was 108.86 ± 1.13 nm, the potential was −21.75 ± 2.36 mV, and the PDI was 0.297 ± 0.006, which indicated the good reproducibility of the process. The variance analysis is shown in Appendix A, and the obtained value of *p* < 0.05 indicates that the model is significant.

### 2.2. Preparation of PTG and PSG

According to the single-factor experiment results, the composition of PTG in the prescription should be 1% carbomer, 17% glycerol, and 0.5% triethanolamine, and the content of phloretin should be 1.22 mg/g. To prepare the gels, appropriate amounts of Phl-TFs were weighed and dissolved in a carbomer matrix overnight. Then, appropriate amounts of glycerin and triethanolamine drops were alternately added to adjust the pH value. In the PSG, the Phl-TFs were replaced with the Phl suspension. The prepared gels were milky white, delicate, and easy to spread with good adhesion and uniformity, meeting the quality standards of gels.

### 2.3. Characterization of Phl-TFs

The EE of the Phl-TFs was 88.02% ± 1.92% (*n* = 3, $\bar{x}$ ± SD), and the DL was 8.06% ± 0.10% (*n* = 3, $\bar{x}$ ± SD). The average particle size was 108.86 ± 1.13 nm (Figure 1A), and the zeta potential was −21.75 ± 2.36 mV. (*n* = 3, $\bar{x}$ ± SD) (Figure 1B).

The TEM images (Figure 1C) showed that the Phl-TFs exhibited a circular or elliptical-like surface, and the observed dimensions were close to those determined using the Malvern nanoparticle size analyzer.

In the DSC thermogram (Figure 1D) of the Phl-TFs, a clear absorption peak was observed at approximately 269 °C, which indicated that there was a weak interaction between the components in the physical mixture. Phl-TFs existed in the molecular or amorphous state.

The XRD pattern of the Phl-TFs (Figure 1E) exhibited a strong diffraction peak at 13. 8944°, and similarly, the PM showed a peak at the response to this position, suggesting that the crystalline form in the mixture is the same as that of phl, which is consistent with the results of the DSC evaluation.

The -OH vibration appears at 3359 cm^−1^, and the characteristic absorption peak of phl is at 1638.39 cm^−1^, which is the C=O stretching vibration (Figure 1F). The spectrum of the PM shows a simple superposition of these characteristic peaks, while in the spectrum of the phl powder, the C=O stretching vibration peak is significantly weakened, suggesting its entry into the nanovesicular carrier formed by the surfactant and phospholipid bilayer.

A Log P > 1 indicates better lipid solubility, which facilitates absorption. The experimental results (Table 1) showed that the oil–water partition coefficient of the Phl-TFs was greater than 2, which indicated that the drug was more lipid-soluble and less water-soluble; compared with Phl, for Phl-TFs, the oil–water distribution coefficient was significantly improved, and the permeability increased significantly (*p* < 0.05, paired *t* test).

The stability test results are shown in Figure 2. The Phl-TF formulations were essentially stable on Days 7, 14, and 21 compared to Day 0 with no significant differences in the appearance or property characterization parameters (*p* > 0.05, paired *t* test). Thus, the optimized prescription formulations have good stability when stored at 4 ± 2 °C and 25 ± 2 °C for 21 days.

### 2.4. In Vitro Skin Permeation Studies

#### 2.4.1. Transcutaneous Infiltration

The data from the in vitro skin permeation experiments are shown in Figure 3A. The cumulative permeation of Phl over 24 h was the smallest at 442.57 ± 44.71 μg/cm^2^, whereas the cumulative permeation of Phl-TFs was the largest at 842.73 ± 20.86 μg/cm^2^, due to the slow release effect of the gel itself. Moreover, the cumulative permeation of PTG was smaller than that of Phl-TFs at 615.31 ± 44.32 μg/cm^2^. Compared with Phl, both the Phl-TFs and PTG significantly improved the skin permeability of Phl (*p* < 0.05, paired *t* test).

#### 2.4.2. Skin Retention

The data from the in vitro skin retention experiments are shown in Figure 3B. The skin retention at 24 h was 377.84 ± 26.26 μg/cm^2^ for the Phl, 536.05 ± 35.56 μg/cm^2^ for Phl-TFs, and 1086.27 ± 15.27 μg/cm^2^ for PTG. Thus, phloretin retention in the skin was significantly better with PTG than the Phl-TFs (*p* < 0.05, paired *t* test).

### 2.5. Skin Irritation Experiment

After visual inspection of the skin of rats to which Phl-TFs had been applied, no redness or swelling was observed. Microscopic images of skin cross-sections did not show any significant differences between the blank group (Figure 3C(a)) and the Phl-TF group (Figure 3C(b)) or between the PTG (Figure 3D(b)) and the blank group (Figure 3D(a)). Therefore, it can be concluded that the prepared PTG is safe and nonirritating when applied to the skin.

### 2.6. In Vivo Pharmacokinetics

After PSG and PTG had been applied to the backs of rats, samples were taken at different time points to determine the concentration of Phl in plasma and to investigate the pharmacokinetic characteristics of PSG and PTG.

Sprague–Dawley (SD) rats were percutaneously administered 100 mg/kg PTG or 100 mg/kg PSG. The mean blood-concentration–time curves are shown in Figure 4, and the main pharmacokinetic parameters are shown in Table 2. The half-life (T_1/2_) of PSG was approximately 7.65 h, with a peak time of approximately 2.13 h and a mean residence time (MRT) of approximately 9.99 h. Compared with PSG administration, PTG increased the C_max_ and T_1/2_ in rats by 0.85 μg/mL and 8.06 h, respectively. In addition, the AUC_0_-∞ of PTG was 8.22 μg/mL·h, which was significantly 1.97 t higher than that in the PSG group (*p* < 0.05, paired *t* test), indicating that PTG could improve the bioavailability of Phl.

### 2.7. Antioxidant Activity of PTG in a Rat Model of Aging Induced by D-gal

#### 2.7.1. Morris Water Maze (MWM)

The Morris water maze test was used to evaluate the spatial memory of the rats in each group. In the directed navigation experiment, the escape latency of all groups of rats showed a gradual decrease. On the first day of training, the swimming trajectories of the rats in each group were very disorganized and irregular; by the fifth day, the escape latencies in the Vitamin E, PSG, PTG-H, and PTG-M groups were significantly shortened (Figure 5A,B). Compared with the control group, the latency period on the fifth day of the experiment was significantly higher in the model group than in the control group (*p* < 0.05, paired *t* test), indicating that modeling was successful. In addition, the latency period in the PTG-H group was similar to that of the Vitamin E group and significantly lower than that of the model group as shown in Table 3 (*p* < 0.05, paired *t* test). As shown in Figure 5C, after the platform was removed on the last day of the experiment, the numbers of platform crossings in the Vitamin E group and the high-dose PTG-H group were significantly greater than that in the model group (*p* < 0.05, paired *t* test). For the other PSG and PTG groups, although the number of platform crossings increased compared with the model group, the difference was not significant.

#### 2.7.2. Organ Indices

The kidney, liver, and thymus indices in the model group were significantly smaller than those in the control group (*p* < 0.05, paired *t* test). Compared with the model group, the Vitamin E group showed significantly increased rat organ indices, and the PTG groups demonstrated increased liver and thymus indices, among which the liver and thymus indices in the PTG-H group were significantly increased (Figure 6) (*p* < 0.01, paired *t* test).

#### 2.7.3. CAT, GSH-Px, MDA, SOD, and Content of T-AOC in Aging Rats

##### Content in the Serum

As shown in Figure 7, compared with the control group, the activities of CAT and GSH-Px were highly significantly decreased in the model group (*p* < 0.01, paired *t* test), and the activities of SOD and T-AOC were significantly decreased (*p* < 0.05, paired *t* test). Moreover, the MDA content in the model group was significantly higher than that in the control group (*p* < 0.05, paired *t* test). Compared with the model group, the vitamin E group displayed significantly increased activities of CAT, SOD, and GSH-Px in serum and a decreased MDA content. The data showed that PSG significantly increased the activities of CAT, SOD, GSH-Px, and T-AOC in the serum of rats compared with the model group (*p* < 0.05, paired *t* test). Compared with the model group, the activities of CAT, SOD, T-AOC, and GSH-Px were significantly increased in the PTG group, while the content of MDA was significantly decreased (*p* < 0.05, paired *t* test). The enzymatic activities of SOD and GSH-Px were extremely significantly increased in the high-dose group (*p* < 0.01, paired *t* test).

##### Content in the Skin Tissue

As shown in Figure 8, CAT, SOD, GSH-Px, and T-AOC activities in the skin tissues of the rats in the PTG groups were increased, among which SOD activity in the high-dose group and GSH-Px activity in the high- and medium-dose groups were significantly increased compared with those in the model group (*p* < 0.01, paired *t* test). Additionally, the MDA content in the high-dose group was significantly decreased (*p* < 0.01, paired *t* test), while the other doses significantly increased the activities of CAT, SOD, GSH-Px, and T-AOC and decreased the content of MDA (*p* < 0.05, paired *t* test).

## 3. Discussion

Despite its multiple biological activities, the application of phl is limited by its photoinstability and poor water solubility [19]. Moreover, little research has been reported on the skin-specific effects of phl [20,21]. Currently, the primary transdermal drug delivery formulations of phl are solid dispersions and microemulsion gels [22]. TFs are effective transdermal carriers that can help hydrophobic drugs dissolve faster and encapsulate them more effectively [23]. Despite the fact that TFs can increase the limited bioavailability of phl, research on the appropriate prescription dosage and in vivo tests to measure their antioxidant efficacy are still lacking [24]. In this study, we optimized the formulation to prepare Phl-TFs. The resulting Phl-TFs were characterized in terms of their physical and morphological properties and underwent in vitro tests for skin penetration, stability, and irritation tests in vitro.

To ensure that active ingredients penetrate the skin, the particle sizes of powders used in skin care products must be monitored and controlled [25]. Formulation ratios can be optimized using zeta potential measurements to reduce aggregation, which aids in the penetration of active ingredients into the dermis [26]. There are reports that the higher the zeta potential is, the more stable the vesicles are. At the optimized lecithin-to-cholesterol ratio of 2.67:1, the Phl-TFs had a stable and very small particle size. This conclusion is consistent with that of Binghui Chen et al. [27]. The ZP of the Phl-TFs in this study was −21.75 ± 2.36 mV, allowing it to be considered a stable prescription. According to the TEM morphology analysis, the surface of the Phl-TFs was round or elliptical, and strong particle aggregation was observed. This may be due to the presence of amorphous phl powder in the TFs. FT-IR spectra of Phl, SPC, CHOL, SDC, physical mixture (PM), and Phl-TF powder FT-IR spectra were compared for structural characterization, revealing a characteristic absorption peak of phl at 1638.39 cm^−1^ and a noticeably weaker C=O stretching vibration absorption peak from Phl-TFs. These data were supported by DSC, which displayed a heat absorption peak and indicated an amorphous form. Phl produced a significant diffraction peak in the comparative XRD pattern at 13.8944° (A), and the physical mixture (E) displayed a peak at this same position, indicating that the crystalline form in the mixture is the same as that in the phl powder.

Previous studies have demonstrated that TFs can transport intact encapsulated drugs to the deeper layers of the skin or bloodstream through the trans-epidermal permeation gradient in the stratum corneum [28]. The results of in vitro permeation and skin retention assays revealed that Phl-TFs have a higher cumulative permeability compared to PTG. This difference could be attributed to the reticular structure of the gel, which delays drug release [29]. Based on the stability and in vitro permeability of the evaluated Phl-TFs, those prepared with the optimal formulation showed stable and durable absorption by rats compared with the other formulations. The authors investigated the pharmacokinetic experiments of phloretin and Phl-TFs in SD rats after transdermal administration. Pharmacokinetic tests were carried out with phl in the laboratory for the trial. Phloretin’s T_1/2_ in rats after intravenous administration of 20 mg/kg was 7.569 h [30]. When phl was given transdermally at the same dose as that in the Phl-TF group, the rats’ C_max_ and T_1/2_ rose by 0.85 g/mL and 8.06 h, respectively. The time required to eliminate 63.2% of the drug from the body is defined as the mean retention time (MRT), which also indicates the average amount of time the drug spends in the body [31]. The experimental results showed that the MRT in the PTG group was greater than that in the PSG group. This difference was most likely because it takes time for phl to be released from the vesicle carrier; consequently, there is a gradual release effect.

The D-galactose-based aging model has gained widespread acceptance as an animal model for both brain aging and application aging [32]. The cognitive abilities of the rats were assessed using the Morris water maze experiment, which was designed to measure the animals’ learning and memory abilities [33]. Extra oxygen radicals produced by the metabolism of D-galactose boost the activity of peroxidation products while decreasing the activity of enzymes that scavenge free radicals in brain-damaging tissues [34]. During oxidative stress, SOD transforms a significant quantity of xanthine dehydrogenase to xanthine oxidase, speeding up the breakdown of cellular structures and organism aging [35]. The marker enzyme of peroxidase is CAT [36]. The antioxidant enzyme GSH-Px plays a crucial role in the body’s defense against free radicals [37]. The body’s overall antioxidant capacity is represented by T-AOC [38]. As a byproduct of lipid peroxidation, MDA can be used to indirectly measure the degree of lipid peroxidation in the body and the harm that oxygen radicals have caused to cells [39]. CAT in rat serum increased dramatically in the PTG-H group, indicating that their skin permeability had effectively increased. In addition, both PTG groups displayed dramatically increased GSH-Px activity in skin tissue homogenates compared to the model group. GSH-Px activity was not significantly higher in the PSG group. This could be explained by the low water solubility of Phl, which would mean that the generated Phl-TFs were able to penetrate through the stratum corneum into the dermis, where they would then enter the bloodstream [40]. Previous studies have shown that different doses of lycopene and chlorpromazine cause a significant decrease in the apoptosis rate and a significant increase in serum CAT, GSH-Px, and SOD levels in hippocampal tissues, thus reducing oxidative stress and neurological damage [41], which is consistent with the results of our study.

The study showed that Phl-TFs improved the learning and memory capacity, as well as the antioxidant function, in D-galactose-induced aging model rats. Previous research has demonstrated that applying UV irradiation to mouse skin elicits a stress response, resulting in hippocampal and cutaneous nerve damage [42]. As a consequence, skin aging can lead to hypothalamic and hippocampal dysfunction, ultimately causing decreasing body function and causing aging [43]. The topical application of antioxidant drugs to the skin can be considered when developing new anti-aging techniques that slow down aging in organisms.

## 4. Materials and Methods

### 4.1. Materials

SPC, cholesterol (CHOL), Sodium deoxycholate (SDC), Soy lecithin (purity ≥ 98%, lot no. J27M8R32632, Shanghai Maclean Biochemical Technology Co., Ltd., Shanghai, China), egg yolk lecithin (purity ≥ 98%, lot no. T21O8F46085, Shanghai Yuanye Biotechnology Co., Ltd., Shanghai, China), methanol and acetonitrile (Thermo Fisher Scientific, Waltham, MA, USA); other chemical reagents were all analytical-grade reagents (Tianjin Tianli Chemical Reagent Co., Ltd., Tianjin, China); PTG (batch number: 20201128, laboratory homemade), carbomer 934 (batch no.: 20200526, Langfang Jutong Chemical Co., Ltd., Yongqing, China), sodium carboxymethyl cellulose (CMC-Na, Tianjin Comio Chemical Reagent Co., Ltd., Tianjin, China), hydroxypropyl methyl cellulose (HPMC, batch no.: 20180115, Tianjin Comio Chemical Reagent Co., Ltd.), triethanolamine (batch no.: 20170708, Tianjin Tianli Chemical Reagent Co.), Glycerol (Batch No.: 20170815, Tianjin Tianli Chemical Reagent Co., Ltd.). PSG (Batch No.: 20201205, Laboratory Self-made); D-Galactose (Purity ≥ 98%, Batch No.: R24O11H128688, Shanghai Yuanye Biotechnology Co., Ltd.), Vitamin E soft capsule (Batch No. 20200810, Zhejiang Pharmaceutical Co., Ltd., Xianju, China), Oxide dismutase (SOD) test kit (A001-3), Malondialdehyde (MDA) test kit (A003-1), Glutathione peroxidase (GSH-Px) test kit (A005-1), Total Antioxidant Capacity (T-AOC) Assay Kit (A015-2-1), Protein Quantification (TP) Assay Kit (A045-2), Catalase CAT Test Kit (A007-1-1) (Nanjing Jiancheng Biological Research Institute, Nanjing, China)

### 4.2. Animals

SPF male SD rats, body weight 180~220 g, were purchased from Chengdu Dasuo Laboratory Animal Co., Ltd. (Chengdu, China). All of the animals underwent a week of acclimatization with a room temperature of (25 ± 2) °C, relative humidity in the range of (50 ± 10)%, and circadian rhythmic lighting. The entire process complied with the ethical standards for using animals. The Shaanxi University of Traditional Chinese Medicine Ethics Committee granted approval for the animal experimentation protocols (Approval No. SUCMDL20200915001).

### 4.3. Preparation of the Phl-TFs and Optimization of the Formulation

#### 4.3.1. HPLC Analysis

HPLC (LC-16 Shimadzu, Kyoto, Japan) was used to determine the concentrations of phl. A reversed-phase C18 column (5 μm, 4.6 × 250 mm) was used with mobile phases A (acetonitrile) and B (0.1%-Phosphoric acid) eluted equally. The flow rate was 0.8 mL/min, the column temperature was 30 °C, and the injection volume was 10 μL. Phl was detected at a wavelength of 280 nm. The results of the HPLC methodological investigations (specificity, standard curve, precision, stability, repeatability, and sample recovery rate) can be found in the Appendix A.

#### 4.3.2. Preparation of the Phl-TFs

The Phl-TFs were prepared by using the thin-film hydration-ultrasound method. SPC (200 mg), CHOL (120 mg), phloretin (10 mg), and sodium deoxycholate (30 mg) were dissolved in 30 mL of anhydrous ethanol, and the mixture was subjected to rotary evaporation at 45 °C to remove the anhydrous ethanol and form a uniform membrane. After adding 10 mL of PBS and hydrating the mixture at 45 °C for 1 h, the suspension was transferred. Phl-TFs were created using an ice-water bath and probe ultrasonography for five minutes at 150 W (working 5 s, stopping 2 s), and a 0.45 μm microporous filter membrane filtration was obtained.

##### Single-Factor Experiments

Single-factor experiments were carried out to evaluate the effects of lecithin type (SPC), egg yolk lecithin, natural phospholipids, SPC to CHOL ratio (1:1, 2:1, 4:1, 6:1, 8:1, and 10:1 *w*/*v* at a concentration of SPC of 20 mg/mL), phloretin concentration (0.5, 1, 2, 3, 4, and 5 mg/mL), hydration time (15, 30, 45, 60, 75, and 90 min) and surfactant content (0, 5, 10, 15, 20, and 30%) on the Phl-TFs. The prescription was considered improved using the index encapsulation efficiency (EE).

##### Box–Behnken Response Surface Method to Optimize Prescription

Based on the results of the single-factor experiments, the formulation of phloretin was optimized using the Box-Behnken design process with Design Expert (version 8.0.6). A (SPC/CHOL, *w*/*v*%), B (phloretin concentration, mg/mL), and C (hydration time, min) were established in the ranges of 1~4%, 0.5~2 mg/mL, and 15~75 min, respectively, for the experiments. To compare the Phl characteristics, the response variable EE (%; Y) of the TFs was selected. To determine the model fitting and evaluate the mathematical relationship between the independent and response variables, the design contained 17 experimental conditions.

##### Factorial Design for Prescription Optimization

The Phl-TF preparation procedure was optimized by considering the findings from the prior single-factor experiments. The ultrasound time (min; A) and ultrasound power (W; B) were taken as independent variables, and based on the preliminary experimental results, three levels of 4~5 min and 135~225 W were selected for the experiments in permutations. Each set of experiments was performed three times in parallel. The assessment indices used to compute the OD value and determine the ideal preparation conditions were vesicle size (VS), polydispersity index (PDI), zeta potential (ZP), and EE.
$$\mathrm{dimax}=\frac{\mathrm{Yi}-\mathrm{Ymin}}{\mathrm{Ymax}-\mathrm{Ymin}}$$
$$\mathrm{dimin}=\frac{\mathrm{Ymax}-\mathrm{Yi}}{\mathrm{Ymax}-\mathrm{Ymin}}$$
$$\mathrm{OD}={{(d}_{1}d_{2}d_{3}\ldots )}^{1/k}$$

Ymax value is its maximum; Ymin is its minimum; and *k* is the number of indicators.

### 4.4. Characterization of the Phl-TFs

#### 4.4.1. Transmission Electron Microscopy (TEM)

The Phl-TFs were prepared according to the optimal formula and procedure and diluted with the appropriate amount of PBS. Then, 200 μL of the diluted solution was added dropwise to a copper grid covered with carbon film, and after standing for one minute, the sample was stained with 2% tungsten phosphate solution and dried. Transmission electron microscopy (TEM) was used to view the morphology of the vesicles.

#### 4.4.2. Analysis of Particle Size and Zeta Potential

Using a nanoparticle analyzer (ZS90, Malvern, UK), the VS, PDI, and ZP were examined during the preparation of the best Phl-TFs. Each sample was evaluated at 25 °C with 3 parallel samples.

#### 4.4.3. Encapsulation Efficiency (EE%) and Drug Loading (DL%)

Precisely 1 mL of Phl-TFs was added to a 10 mL volumetric flask followed by methanol, and the sample was shaken well and sonicated for 15 min. The sample was then filtered through a 0.22 μm membrane using ultrafiltration centrifugation to separate the Phl-TFs from Phl. HPLC was used to determine the drug content. Finally, the sample was centrifuged, and the EE% and DL% were calculated.
$$\mathrm{EE}(\%)\;=\frac{W_{\mathrm{total}}-W_{\mathrm{free}}}{W_{\mathrm{total}}}\times 100\%$$
$$\mathrm{DL}\;(\%)=\frac{W_{\mathrm{total}}-W_{\mathrm{free}}}{W_{\mathrm{lipal}}}\times 100\%$$

The terms “W_total_”, “W_free_”, and “W_lipal_” refer to the total weights in the assay and the amounts of free and total drugs in the Phl-TFs, respectively.

#### 4.4.4. Differential Scanning Calorimetry (DSC)

DSC was used to investigate the physical condition of the drug within the Phl-TF vesicles. To investigate the thermal properties of phl, CHOL, SPC, SDC, physical mixture (PM), and lyophilized powder of Phl-TFs, the Phl-TFs were rapidly precooled in a −80 °C cryogenic freezer and then freeze-dried in a freeze dryer (−50 °C, −0.01 MPa) for 24 h to obtain the Phl-TF lyophilized powder. The samples were heated at a rate of 5 °C·min^−1^ under a constant nitrogen flow in a temperature range of 10~400 °C.

#### 4.4.5. X-ray Diffraction (XRD)

With a Bruker AXS D8 diffractometer, several samples (Phl, CHOL, SPC, SDC, the PM, and the Phl-TF lyophilized powder) were scanned using a Cu-K radiation source fixed at 40 kV and 100 mA. Data were collected using scattering angle (2θ) ranging from 10° to 70°, and the scan speed was 1°/s.

#### 4.4.6. Fourier Transform Infrared (FTIR) Spectroscopy

Studies to evaluate the compatibility of the drugs and excipients were carried out utilizing FT-IR and the KBr compression method. The samples were vacuum-dried at 35 °C for 24 h. Appropriate amounts of KBr were ground through a sieve, dried at 105 °C for 24 h and set aside. Phl, CHOL, SPC, SDC, physical mixture (PM), and the Phl-TF lyophilized powder were vacuum-dried at 35 °C for 24 h. KBr and samples were accurately weighed, and KBr was mixed with the sample at a ratio (200:1 *w*/*w*). After being properly weighed and compressed, the samples were analyzed using an infrared spectrophotometer (TENSOR-27, Bruker, Germany) in the wavelength range of 4000~400 cm^−1^.

#### 4.4.7. Determination of the Oil–Water Partition Coefficient

Certain amounts of n-octanol and water were weighed and added to a conical flask. The samples were agitated magnetically at 25 °C for 48 h and then allowed to stand overnight in a separatory funnel. Separately, 2 mL of Phl solution and Phl-TF aqueous solution (C_0_) were combined and then agitated in a water bath in the dark at 37 °C for 24 h. A sample of 0.5 mL of the separated solution (C_w_) was obtained after separation at 4000 RPM for 10 min, after which it was diluted with 1.5 mL of methanol and then filtered through a 0.22 μm membrane. Finally, the supernatant was analyzed by HPLC. The oil–water partition coefficient (Papp) and Ig Papp were then calculated.
$$P_{\mathrm{app}}=\frac{C_{w}}{\left(C_{0}-C_{w}\right)}$$

### 4.5. Stability Study

The samples were stored at 4 °C and 25 °C for 21 days to evaluate the stability of the selected Phl-TF formulation and to determine the EE, ZP, VS, and PDI at fixed time intervals (0, 7, 14, and 21 days).

### 4.6. In Vitro Permeation of the Phl-TFs

#### 4.6.1. Permeation through Rat Skin

Skin permeation was evaluated using an improved Franz diffusion cell at (37 ± 0.2) °C; the permeability of the optimized Phl-TFs (1 mg·mL^−1^) was compared with that of the phloretin suspension. The fascia and subcutaneous fat of the rat abdomen were removed, and a 2 × 2 cm^2^ piece of skin was obtained. The skin was placed between the donor and recipient chambers with a diffusion area of 0.785 cm^2^ and stirring at (300 ± 5) r·min^−1^, using PBS (pH = 7.4) as the recipient medium. At predetermined time points (0.5, 1, 2, 4, 6, 8, 12, and 24 h), samples (0.5 mL) were removed from the recipient chamber, and the medium was immediately replaced with fresh PBS.
$$Q_{n}=\frac{C_{n}\times V_{o}+\sum_{i=1}^{n-1}C_{i}\times V_{i}}{S}$$

$C_{n}$ is the concentration of phloretin in the receptor chamber at hour n, $C_{i}$ is the drug concentration of the sample, S is the effective diffusion area (0.785 cm^2^), $V_{o}$ is the total volume of the test solution (15 mL), and $V_{i}$ is the sampling volume (0.5 mL).

#### 4.6.2. Skin Retention Study

After the permeation experiment, the skin surface was cleaned with saline and wiped with filter paper to remove the residual drug. Then, the filter paper was cut into small pieces, sonicated with 1.0 mL of methanol for 10 min, extracted three times, and centrifuged at 12,000 r·min^−1^ for 10 min, after which the supernatant was collected. This procedure was performed to estimate drug retention in the skin. The drug content in the skin was measured by using the validated HPLC method after the extraction solution was filtered through a 0.45 μm filter membrane. The cumulative amount of phl on the skin per square centimeter (Q_n_) was calculated according to the following equation.

#### 4.6.3. Skin Irritation

Irritation to SD rat skin caused by the formulation was evaluated by HE staining. Six SD rats were randomly divided into a blank control group and a Phl-TF group. In the Phl-TF group, the formulation was uniformly applied to the skin on the back of the rats, covered with gauze, and fixed with medical tape for 24 h. After the rats were euthanized, the treated skin areas were excised and fixed in 4% paraformaldehyde solution for 24 h, and the tissues were embedded, sectioned, and stained for observation under a microscope.

### 4.7. In Vivo Pharmacokinetic Studies

Prior to the experiment, male SD rats were fasted for 12 h fast before being randomly separated into 2 groups: the PSG group and PTG group (100 mg/kg), each with six rats. After treatment, blood was drawn from the orbit at 0, 0.5, 1, 2, 4, 6, 8, 10, and 24 h into an anticoagulant centrifuge tube containing 1.5 mL of heparin sodium. Appropriate samples were added to methanol, swirled for 2 min, and cryocentrifuged at high speed (15 min, 4 °C, 13,500 rpm). The supernatant was removed and stored at −20 °C, and 200 μL plasma sample was added to 600 μL of methanol to precipitate the protein, followed by vortexing for 2 min, freezing, and centrifugation at high speed for 15 min. The supernatant was passed through a 0.22 μm organic microporous filter membrane, and HPLC was used for further analysis. Data from these samples were used to construct the pharmacokinetic profiles by plotting the concentration of phloretin in plasma vs. time.

### 4.8. PTG Antioxidant Efficacy in D-gal-Induced Aging Rats

#### 4.8.1. D-gal Model of Aging Rats

SD rats were randomized into 7 groups and adaptively fed for one week accordingly: a blank control group, a D-galactose model group, a Vitamin E group (26.99 mg/kg), a PSG group (100 mg/kg), a PTG-L group (50 mg/kg), a PTG-M group (100 mg/kg), and a PTG-H group (200 mg/kg). The drug was applied to the dorsal skins of the rats. The rats’ backs were treated with saline in the model group and the blank control group. Except for the blank control group, which received the same dose of regular saline instead of D-galactose, the rats in each group received intraperitoneal injections of D-galactose (200 mg/kg). The experimental duration was 8 weeks.

#### 4.8.2. Morris Water Maze (MWM)

The Morris water maze test was started after drug administration. SD rats were placed sequentially at the entrances of the four quadrants for 5 consecutive days, while a video tracking system was used to record the escape latency. The escape latency of rats that failed to enter the platform was taken as 90 s. Training was conducted once a day for 5 d. The mean result was calculated and the swimming trajectory of each group was recorded. On the 6th day, the hidden platform was removed, and the spatial exploration experiment was conducted. The trajectory of spatial exploration was recorded for each rat for 120 s, and the percentage of time spent entering the platform quadrant and the number of times they entered the platform quadrant were recorded.

#### 4.8.3. Organ Indices

At the end of treatment, the rats were quickly necropsied, the major organs (thymus, liver, and kidneys), fat, and fascia were removed, and the surface fluid was blotted with filter paper. Then, the corresponding organ indices were calculated.

#### 4.8.4. Enzyme Immunoassay Assay of CAT, GSH-Px, MDA, SOD, and T-AOC

The protein concentration of serum and skin tissue samples was determined by using the Coomassie Brilliant Blue method. The collected serum and skin tissues were used with the reagents of the kit to determine the activities of CAT, GSH-Px, MDA, and SOD and the T-AOC content. All steps were performed according to the kit instructions. The absorbance value (A) of each index was determined, and results were calculated according to the formula provided in the instruction manual.

### 4.9. Statistical Analysis

All experimental data are expressed as $\bar{x}$±SD and were evaluated and analyzed using SPSS 19.0 (SPSS Inc., Chicago, IL, USA), DAS 3.2, and GraphPad Prism 8.0 image software. A value of *p* < 0.05 was considered to indicate statistical significance.

## 5. Conclusions

In this study, we successfully prepared Phl-TFs and then proceeded to prepare the PTG. The results of the in vitro permeation assays and pharmacokinetic experiments demonstrated that PTG improves drug penetration into the skin. Moreover, in vivo studies demonstrated that PTG effectively increased the antioxidant enzyme activity and decreased the serum and skin tissue lipid peroxide MDA content in D-gal-induced aging rats. Although PTG plays an important role in antioxidation and anti-aging, developing new formulations for its transdermal delivery is valuable, and its effect on the drug process in vivo has not been clarified and requires further study.

Phl exhibits a diverse range of biological activities and possesses immense potential in the pharmaceutical, food, and cosmetic industries. Its antioxidant and anti-inflammatory properties render it effective in preventing photoaging and treating numerous skin disorders, such as melasma and acne. In the future, the development of Phl should focus on augmenting its efficacy and expanding its scope of applications. The Phl-TFs utilized in this experiment are a topical nanotechnology-based formulation. However, their effectiveness and safety still require clinical confirmation, and their various biological properties for dermatological and cosmetic industry applications merit further investigation.

## Figures and Tables

**Figure 1 molecules-28-06790-f001:**
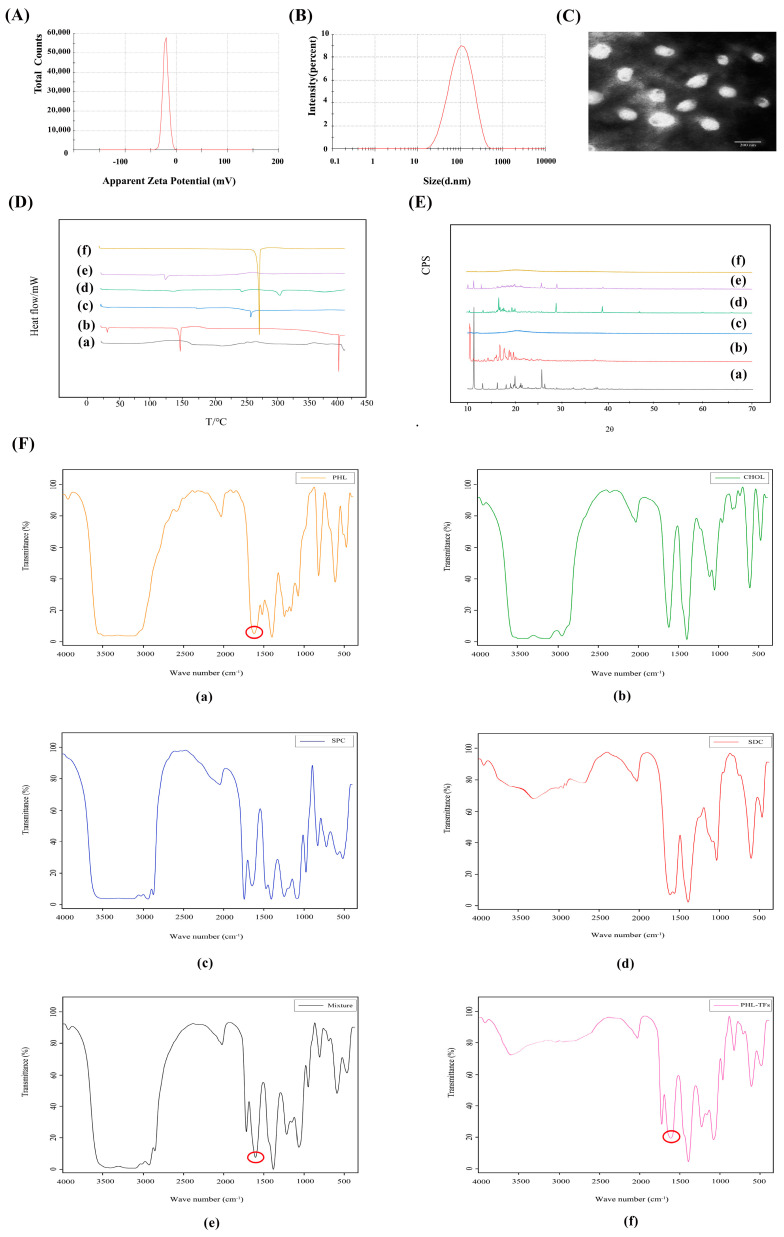
Characterization for Phl−TFs. (**A**) Particle size. (**B**) Zeta potential. (**C**) TEM morphology of Phl−TFs. (**D**) DSC of Phl−TFs. (**E**) XRD of Phl−TFs. (**F**) FTIR of Phl−TFs. Note: (**a**) phl; (**b**) CHOL; (**c**) SPC; (**d**) SDC; (**e**) PM; (**f**) Phl−TFs. Red circles indicate phl absorption peaks.

**Figure 2 molecules-28-06790-f002:**
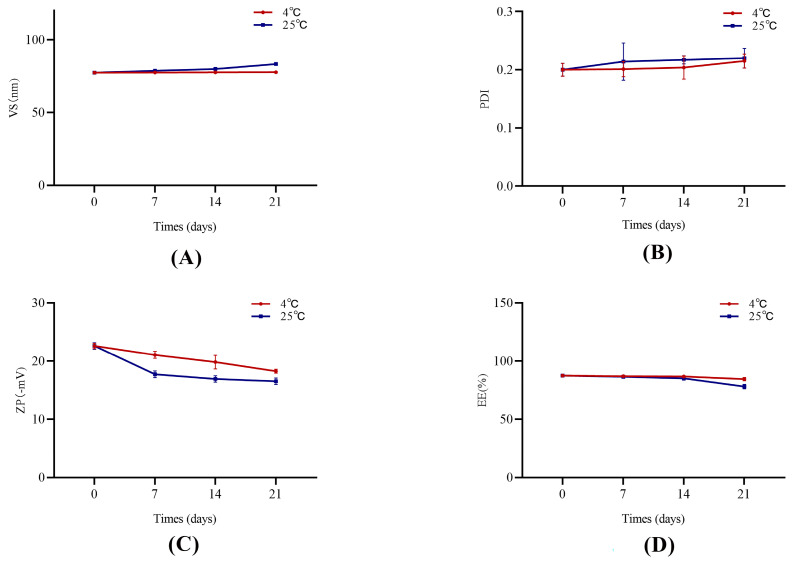
The stability of Phl-TFs at different temperatures. (*n* = 3, $\bar{x}$ ± SD). Note: (**A**) VS, (**B**) PDI, (**C**) ZP, (**D**) EE.

**Figure 3 molecules-28-06790-f003:**
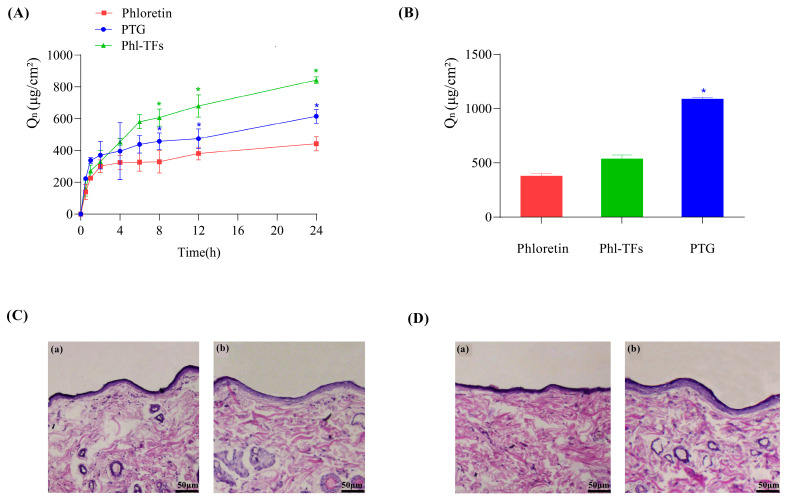
Infiltration and skin retention. (**A**) Cumulative permeation–time curves in vitro (*n* = 3), comparison with phloretin. (**B**) Skin retention volume (*n* = 3), comparison with phloretin. (**C**,**D**) Skin cross-section observed under light microscope after hematoxylin-eosin staining (×200). Note: (**a**,**b**) Compared with the phloretin, * *p* < 0.05. (**C**) (**a**) Phl-TF blank group. (**b**) Phl-TF group. (**D**) (**a**) PTG blank group. (**b**) PTG group.

**Figure 4 molecules-28-06790-f004:**
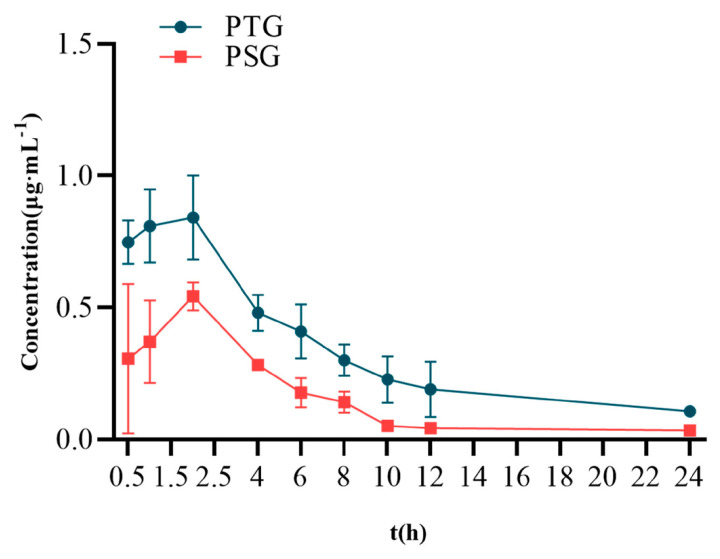
Mean blood−concentration−time curve (*n* = 6, $\bar{x}$ ± SD).

**Figure 5 molecules-28-06790-f005:**
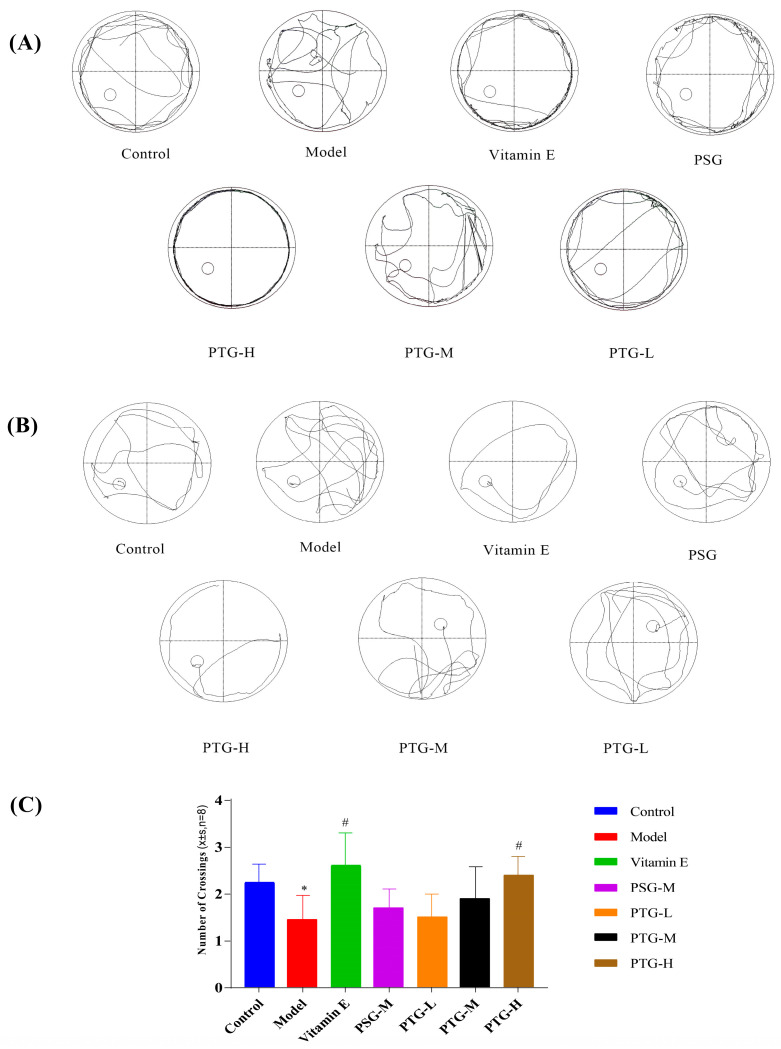
Swimming trajectory of rats. (**A**) Day one of the orientation navigation experiment. (**B**) Day five of the orientation navigation experiment. (**C**) Morris water maze rats crossing platform times (*n* = 8, $\bar{x}$ ± SD). Note: Compared with the control group, ** p* < 0.05. Compared with the model group, ^#^
*p* < 0.05.

**Figure 6 molecules-28-06790-f006:**
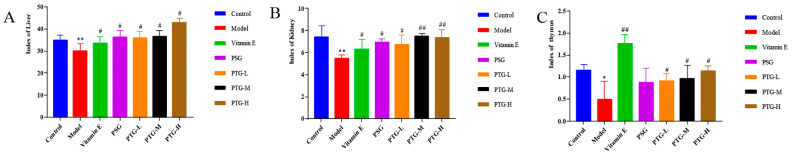
Organ index in aging rats. Note: (**A**) liver; (**B**) kidney; (**C**) thymus. Compared with the control group, * *p* < 0.05, ** *p* < 0.01. Compared with model group, # *p* < 0.05, ## *p* < 0.01.

**Figure 7 molecules-28-06790-f007:**
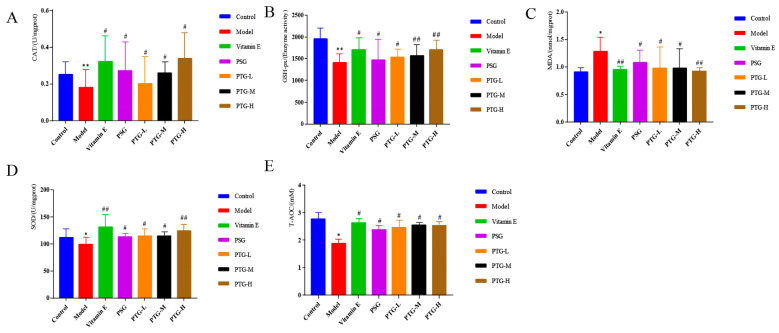
CAT, SOD, GSH-Px, T-AOC, and MDA levels in the serum of aging rats. Note: (**A**) CAT. (**B**) GSH-Px. (**C**) MDA. (**D**) SOD. (**E**) T-AOC. Compared with the control group, * *p* < 0.05, ** *p* < 0.01. Compared with model group, # *p* < 0.05, ## *p* < 0.01.

**Figure 8 molecules-28-06790-f008:**
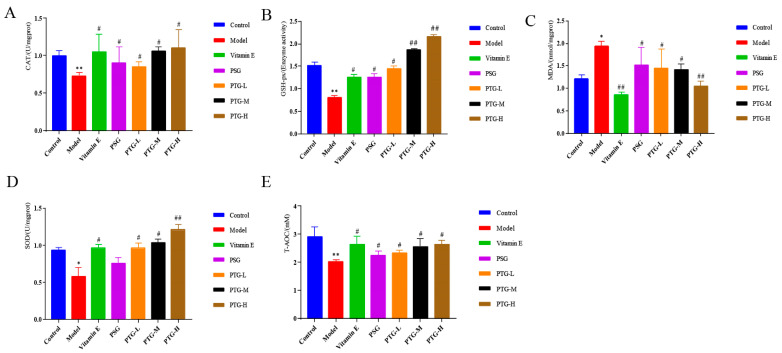
CAT, SOD, GSH-Px, T-AOC, and MDA levels in the skin of aging rats. Note: (**A**) CAT. (**B**) GSH-Px. (**C**) MDA. (**D**) SOD. (**E**) T-AOC. Compared with the control group, * *p* < 0.05, ** *p* < 0.01. Compared with model group, # *p* < 0.05, ## *p* < 0.01.

**Table 1 molecules-28-06790-t001:** Determination of the LogP of Phl-TFs (*n* = 3, $\bar{x}$ ± SD).

Drug	Log P	RSD (%)
Phloretin	1.89	4.92
	1.95	
	2.08	
Phl-TFs	2.26	4.86
	2.37	
	2.15	

**Table 2 molecules-28-06790-t002:** Pharmacokinetic parameters (*n* = 6, $\bar{x}$ ± SD).

Parameter	Unit	PTG	PSG
AUC_(0_–_t)_	μg/mL·h	7.07 ± 1.18 *	3.63 ± 1.71
AUC_(0_–_∞)_	μg/mL·h	8.22 ± 2.15 *	4.18 ± 2.35
MRT_(0_–_t)_	h	7.39 ± 0.76	6.31± 0.98
MRT_(0_–_∞)_	h	11.60 ± 2.53	9.99 ± 2.48
VRT_(0_–_t)_	h^2^	42.12 ± 8.94	36.84 ± 6.22
VRT_(0_–_∞)_	h^2^	162.79 ± 72.02	136.52 ± 48.23
T_1/2_	h	8.06 ± 3.20	7.65 ± 1.20
T_max_	h	1.69 ± 0.59	2.13 ± 0.84
C_max_	μg/mL	0.85 ± 0.06	0.61 ± 0.26

Note: compared with the PSG group, * *p* < 0.05.

**Table 3 molecules-28-06790-t003:** Effect of phloretin on the latency of evasion in rats (*n* = 8, $\bar{x}$ ± SD).

Incubation Period(s)	Day 1	Day 2	Day 3	Day 4	Day 5
Control	85.00 ± 0.01	87.14 ± 12.21	64.15 ± 20.20	45.90 ± 14.25	39.44 ± 13.21
Model	88.59 ± 31.42	88.09 ± 12.23	69.00 ± 14.58	62.20 ± 14.07 *	54.93 ± 10.67 *
Vitamin E	73.80 ± 18.13 ^#^	60.70 ± 13.61 ^#^	48.40 ± 16.50 ^##^	37.10 ± 10.40 ^##^	36.65 ± 11.54 ^#^
PSG	89.27 ± 23.35	83.37 ± 10.33	66.40 ± 16.57	65.00 ± 12.92	55.42 ± 18.03
PTG-H	80.88 ± 6.71 ^#^	80.60 ± 21.38	58.10 ± 10.17 ^#^	55.80 ± 12.48 ^#^	42.90 ± 10.30 ^#^
PTG-M	81.29 ± 27.5	84.35 ± 21.37	59.20 ± 19.10 ^#^	57.70 ± 21.92 ^#^	49.80 ± 5.68 ^#^
PTG-L	85.72 ± 9.18	84.94 ± 10.79	64.10 ± 25.45	57.94 ± 15.05 ^#^	49.75 ± 6.47 ^#^

Note: Compared with the blank group, ** p* < 0.05. Compared with the model group, *^#^ p* < 0.05, *^##^ p* < 0.01.

## Data Availability

Data are contained within the article.

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
