# Peer review of "Phloretin Transfersomes for Transdermal Delivery: Design, Optimization, and In Vivo Evaluation"

_molecules, 2023, doi:10.3390/molecules28196790_

Round 1

Reviewer 1 Report

Authors report about phloretin transfersomes for transdermal delivery. Readers of Molecules are interested in the results. Therefore, this manuscript is worthy being published in Molecules.

1. Page 7, Line 277. Raio of SPC to CHOL was determined 2:1. The encapsulation rate of 2:1 was 94.06% but that of 6:1 was 96.4%. Why 2:1 is better than 6:1?

2. Page 8, line 344. In Figure 1 (6), the peak at 1638.39 is not clear. Please assign the absorption peak.

3. Page 15. Figure 6 is too small. 

Reviewer 2 Report

In this work, the authors highlight the need for study of Phloretin Transfersomes for Transdermal Delivery: Design, Optimization and In Vivo Evaluation. The manuscript is mostly well written. I have some major comments and suggestion(s) below:

1.    The abstract seems to be incoherent in language can be re-structured for proper understanding among readers.

2.    Methodology is too short, should provide more elaborative aspect of need for study.

Line 127 as: The Phl-TFs was created by thin film hydration-ultrasound method: SPC 200 mg. This should be re-written what is the meaning of created as it should be developed or prepared.

Line 186 as: scan range 2 is 10-70°, what is this 2 is it scan range or scan range 2, please eloborate

Line 201 as: Stability study

Why stability studies were conducted as per ICH guidelines

Line 279 as: SPC and CHOL of 2:1 was selected. Why? As 4:1 showed highest encapsulation.

Line 289-90 as: The encapsulation rate increased first and then decreased with the increase of hydration time. Please Justify? Why?

3.    All figures should be of high quality and source images to be submitted in original while submitting revisions. Figure 6 should be provided separately not merged for clear and better understanding.

4. Discussion is too long should provide summative assessment findings.

5. I suggest authors that there should be a separate section after conclusion on Future perspectives of Phloretin Transfersomes for Transdermal Delivery on present findings.

Extensive editing of English language required

Reviewer 3 Report

The manuscript entitled “Phloretin Transfersomes for Transdermal Delivery: Design, Optimization and In Vivo Evaluation” reports the assessment of a gel formulation intended for delivering phloretin through the skin to achieve a systemic anti-aging effect.

Although the topic is interesting, the manuscript has many flaws. In the introduction, the authors do not cite relevant references about the use of transfersomes as transdermal delivery systems. The experimental protocol is unclear and poorly detailed making the described experimental procedures not reproducible. Therefore, the materials and methods section should thoroughly revised and all the experiments should be described reporting all the details commonly specified in the literature. For instance, for HPLC analyses, the precision, accuracy and sensitivity of the method should be reported. In in vitro skin permeation experiments, the characteristics of the diffusion system and experimental conditions (e.g. temperature, stirring speed, diffusion area, volume of the tested sample) should be reported.  

English is poor making the meaning of some sentences incomprehensible. For instance, in the “conclusion” (line 610) what is the meaning of the sentence “However, although PTG plays an important role in antioxidant and  anti-aging in this study and has some reference value for new dosage forms of transdermal drug delivery”? English should be thoroughly revised.

English is poor making the meaning of some sentences incomprehensible.

Round 2

Reviewer 2 Report

The authors have addressed all my queries. I recommend the manuscript for its possible publication

Minor editing of English language required

Reviewer 3 Report

The authors revised the manuscript properly.

Moderate revision